# Barcamps or unconferences as an emerging paradigm in medical education: Insights from a pilot and feasibility mixed methods study

Bernd F. M. Romeike[1]*, Johannes Lang[2], Christoph Stosch[3], Sören Moritz[3], Marianne Behrends[4], Martin Lemos[5], Johanna Mink[6], Daniel Tolks[7]

1 Dean's Office for Student Affairs, Medical Education, University Medical Center, Rostock, Germany, 2 Medical Faculty, Giessen University, Giessen, Germany, 3 Student Deans Office, Medical Faculty of the University of Cologne, Köln, Germany, 4 Peter L. Reichertz Institute for Medical Informatics of TU Braunschweig and Hannover Medical School, Hannover Medical School, Hannover, Germany, 5 Audiovisual Media Center, Medical Faculty, RWTH Aachen University, Aachen, Germany, 6 Department of Primary Care and Health Services Research, Nursing Science and Interprofessional Care, Medical Faculty, Heidelberg University, Heidelberg, Germany, 7 Center for Applied Health Science, Leuphana University Lueneburg, Lueneburg, Germany

* bernd.romeike@med.uni-rostock.de

**Data Availability Statement:** The complete datasets cannot be shared publicly because we have taken great care to maintain the confidentiality

## Abstract

Medical education is experiencing a paradigm shift towards more interactive and collaborative pedagogical approaches. Barcamps, also known as unconferences, offer an interactive, participant-driven learning approach. This study aims to evaluate the feasibility of using barcamps as an educational model in medical education. Furthermore, the possibility of establishing barcamps in medical education as a pedagogical tool is discussed. The study integrates two evaluations to achieve a multifaceted understanding: a pilot study conducted in 2022 and a subsequent feasibility study in 2023. Participation in the barcamps and their evaluations was voluntary. We combined quantitative data, primarily from post-event evaluations, and qualitative data from open-ended survey questions. These methods were designed to capture a broad range of participant experiences and perceptions. The pilot study in 2022 included 11 participants and produced a response rate for the evaluations. The feasibility study in 2023 had 34 participants but a lower evaluation response rate of 53%. Both barcamps were generally positively evaluated by the participants, indicating a high level of satisfaction and perceived value. Regarding the active participation of participants, the wide range of presented topics highlights the adaptability and flexibility of the barcamp model. Attendees at the pilot mentioned a significantly higher previous experience with barcamps. The study suggests that barcamps are generally well received as an interactive and collaborative educational tool in medical education, reflected by high participant evaluation scores. The wide range of topics presented and discussed indicates that barcamps can accommodate diverse educational needs and interests. However, the study also identified areas for improvement, most notably in the structure of topic plans and the overall learning environment. Finally, the principals for barcamps might adapt to other educational methods by adding more interaction, choice, collaboration, communication, critical thinking, creativity, and caring to the learning process.

of our participants' data. Sharing the data in a public repository, even in a de-identified format, could still pose a risk to participant privacy, especially since our dataset is small and the possibility of re-identification is not negligible. However, the data underlying the results presented in the current study are available from the Rostock Ethics Committee (contact via ethik@med.uni-rostock.de), for researchers who meet the criteria for access to confidential data.

**Funding:** The author(s) received no specific funding for this work.

**Competing interests:** The authors have declared that no competing interests exist.

# Introduction

## Background

As a consequence of the paradigm shift from teaching to learning, medical education is undergoing a major change, with the traditional lecture-based model giving way to more interactive and collaborative pedagogical approaches. This shift is driven by the need to better prepare future healthcare professionals for the rapidly evolving landscape of medical practice. A possibility to promote cooperative exchange between various persons are so-called barcamps. Similar events used to be called unconferences [1]. In today's literature, the two terms barcamp and unconference are used interchangeably. We prefer the term barcamp because it emphasizes its uniqueness. A barcamp differs from classical scientific conferences in several ways [2–6]. In short, since many of the participants have little experience with the format, a brief introduction to the method is given at the beginning of the event. Although the general theme of the event is known in advance, the agenda and topics of the event are not predetermined by the organizers, but are largely provided by the participants. Participants are encouraged to actively contribute topics and lead discussions. Based on their interests, so-called "idea providers" note session proposals as a short sentence, a question, or a topic for a timely discussion. These are briefly presented to the audience as a "pitch". The audience votes directly on whether there is enough interest in the proposal. Topics, i.e. the notes with enough interest, are immediately collected in the draft session plan. After the last proposal or pitch, the organizers create the final session plan. The number of sessions is limited only by the available space, rooms, and time slots. Each session is documented so that knowledge can be shared beyond the physical confines of the event for both attendees and non-attendees. Participants are free to leave a session midway if it does not meet their learning needs, promoting a dynamic and flexible learning environment. After one or two sessions, participants reconvene in the audience for a summary, feedback, or reflection.

The first barcamp was probably organized in 2003, but similar conventions date back almost 200 years, pioneered by Alexander von Humboldt [7]. Despite the promising aspects of barcamps, there is only a limited amount of literature on their implementation in medical education [4–6].

Due to the scarce literature and the many unique characteristics of barcamps, such as the participant-driven approach with no pre-defined agenda and a focus on open and horizontal communication among participants, we decided to investigate whether barcamps could be used as part of a traditional medical education conference. Additionally, we discuss if the principals for barcamps could also be transferred to other educational methods in medical education by improving the learning process with more interaction and Joe Ruhl's six Cs: choice, collaboration, communication, critical thinking, creativity, and caring [8].

# Materials and methods

## Study design

This study uses a mixed methods approach, combining both quantitative and qualitative data, to investigate the effectiveness of barcamps in scientific medical education conferences. Two separate evaluations were conducted: a pilot study in 2022 [9] and a feasibility study in 2023. The 2023 barcamp was announced by an editorial [10].

We carefully considered the need to obtain ethical approval or written informed consent. Our decision to proceed without it was based on several key factors. First, the nature of the study was non-invasive, low-risk, and focused solely on gathering feedback on the content and delivery of the barcamp. This type of evaluative research is often considered to be within the

realm of standard educational practice rather than research that would require ethical oversight. In addition, the questionnaire was designed to gather general opinions and experiences. The study neither involves sensitive personal data or issues nor does it pose any psychological, social, or physical risks to the individuals involved. Finally, no minors or other vulnerable persons were involved. Given these circumstances, coupled with established norms and guidelines in educational research, we concluded that the study did not fall within the scope of activities requiring ethical approval or written informed consent. Participants gave verbal informed consent by participating on a voluntary and anonymous basis, which was witnessed and documented by the first (BFMR) and last author (DT).

## The barcamps

The planning of the barcamps was based on the personal experiences of BFMR and DT, who had already attended various barcamps and were familiar with the pertinent literature [2, 3]. Various useful materials were downloaded at https://www.selbstlernen.net/ [11]. The general topic of the barcamp in 2022 was "Challenges for Digital Transformation in Education within the New Normal" [9]. In 2023 the focus was on "Generative AI Tools (GAITs) in Medical Education" [10].

## Participants

Participants of the studies were attendees of the medical education barcamps held in 2022 in Halle/Saale, Germany and 2023 in Osnabrueck, Germany as part of the annual conference of the DACH Association for Medical Education (GMA). The barcamps were announced as separate items in the official conference programs. Apart from word-of-mouth advertising, no other recruitment efforts were made. Participation in the evaluation was voluntary and anonymous.

## Quantitative data analysis

Demographic data included age and gender of the participants.
The evaluation of the barcamps was conducted with three questions using Likert scales.

1. Previous experience with barcamps: poor (0) to very good (5)

2. Overall assessment of the barcamp: poor (0) to amazing (5)

3. Probability of attending future barcamps: not likely (0) to very likely (5)

The participants also indicated their agreement (1) or disagreement (0) with various statements about their experience during the barcamps.

1. I learned from others

2. I networked with others

3. It was entertaining and fun

4. I found a way into the topic of the sessions

5. I shared knowledge and expertise

The mean, median, and standard deviation were calculated for all quantitative variables. Mann-Whitney U tests were conducted to compare the distributions between the two barcamps for each quantitative variable.

## Qualitative data analysis

Participants were asked to provide written responses to the following open-ended questions:

1. What were the positive aspects of the barcamp?

2. What was missing and what could be improved at the barcamp?

3. What topic should be discussed at the next barcamp?

For categorization, the authors followed a systematic approach based on Mayring's method [12]. In summary, the participants' responses were translated, structured and paraphrased by one author (BFMR) to capture the core meanings. All other authors independently reviewed and commented on the content to ensure that condensed thematic aspects were correctly translated and that the essential content was retained.

## Results

### Recruitment of participants

No recruitment efforts were made. Reasons for attending were not recorded. According to individual personal communication, participants attended out of general interest in the topics of the two barcamps, curiosity about the barcamp method.

### Pilot study (2022) Halle/Saale, Germany

The pilot study in 2022 attracted 11 participants, of whom 10 (91%) evaluated the barcamp (Table 1). Participants had a mean age of 47.9 years (31–58 years, SD = 9.9). Seven men and three women attended the barcamp. Participants mentioned a rather good prior experience with barcamps with a mean of 2.1 (SD = 1.7) on a Likert scale ranging from poor (0) to very good (5). The overall assessment of the barcamp yielded a favorable mean score of 4.4 (SD = 0.52) on a scale from poor (0) to amazing (5). Furthermore, the likelihood that participants would attend future barcamps was high, with a mean score of 4.5 (SD = 0.71) on a scale from not likely (0) to very likely (5).

During the pilot study in 2022, nine participants pitched 16 topics. Two participants did not pitch a topic. Six participants pitched two topics. The three topics below were selected for presentation and discussion in two parallel sessions:

1. How can the interactive and collaborative tool https://cryptpad.fr/ [13] be used for the documentation of barcamps and for collaborative documents in general?

2. How can we share open educational resources (OER) in medical education?

3. How can we create closeness in online conferences?

Table 1. Descriptive statistics for the pilot study in Halle (2022).

| Variable | Values | Mean | Median | Standard deviation |
|---|---|---|---|---|
| Age | 31–58 years | 47.9 | 50.5 | 9.90 |
| Gender | 7 male / 3 female | | | |
| Previous barcamp experience | | 2.1 | 2.0 | 1.66 |
| Overall assessment | | 4.4 | 4.0 | 0.52 |
| Probability of attending future barcamps | | 4.5 | 5.0 | 0.71 |

## Feasibility study (2023) Osnabrueck, Germany

The feasibility study in 2023 involved 34 participants of whom 18 (53%) evaluated the barcamp (Table 2). Of these, the mean age was 39.9 years (23–57 years, SD = 12.2). 10/18 (55,6%) participants were men. The previous barcamp experience was much lower than in the pilot study, with a mean score of 0,28 (SD = 0.96). The general assessment of the barcamp was also favorable, with a mean score of 3.94 (SD = 0.85). The probability of attending future barcamps was similarly high, with a mean score of 4.33 (SD = 0.77).

During the feasibility study in 2023, eight participants pitched nine topics. Of these, four topics were chosen to be discussed in two parallel sessions:

1. How can generative artificial intelligence (gAI) support student learning?

2. How can gAI support the teaching of lecturers?

3. How can gAI support the generation or use of flashcards?

4. How can gAI support programmatic assessment?

## Comparison between pilot and feasibility studies with Mann-Whitney U Test

To compare demographic data and participant evaluations and intentions to attend future barcamps in the two studies, Mann-Whitney U tests were used. The results are summarized in Table 3. No significant differences were found in age ($p = 0.075$), gender ($p = 0.479$), overall assessment ($p = 0.167$), and likelihood of attending future barcamps ($p = 0.594$). Participants in the pilot study stated a significantly higher prior experience with barcamps ($p = <0.01$).

## Agreement with the statements

In addition to general evaluation metrics, participants were asked to indicate their agreement with various statements about the barcamp experience. The responses were binary, agreeing (1) or not agreeing (0) with each statement.

**Pilot study (2022). I learned from others**: A high percentage of participants (90%) agreed that they learned from others during the barcamp. **I networked with others**: 70% of the participants felt that they were able to network with others. **It was entertaining and fun**: 80% found the barcamp to be entertaining and fun. **I found a way into the topic of the sessions**: 30% felt that they found a way into the topic. **I shared knowledge and experience**: Like networking, 70% agreed that they could share knowledge and experience.

**Feasibility study (2023). I learned from others**: An even higher percentage (94.4%) agreed that they learned from others. **I networked with others**: However, only 33.3% felt they could network with others, a noticeable decrease compared to the pilot study. **It was entertaining**

**Table 2. Descriptive statistics for the feasibility study in Osnabrueck (2023).**

| Variable | Values | Mean | Median | Standard deviation |
|---|---|---|---|---|
| **Age** | 23–57 years | 39.9 | 39.0 | 12.21 |
| **Gender** | 10 male / 8 female | | | |
| **Previous barcamp experience** | | 0.28 | 0.0 | 0.96 |
| **Overall assessment** | | 3.94 | 4.0 | 0.85 |
| **Probability of attending future barcamps** | | 4.33 | 4.5 | 0.77 |

**Table 3. Mann-Whitney U Test results comparing pilot and feasibility studies.**

| Metric | P-Value |
| --- | --- |
| **Age** | 0.075 |
| **Gender** | 0.479 |
| **Previous barcamp experience** | **0.001** |
| **Overall assessment** | 0.167 |
| **Probability of attending future barcamps** | 0.594 |

**and fun**: About two-thirds (66.7%) found it entertaining and fun. **I found a way into the topic of the sessions**: 50% felt that they found a way into the topic, showing an increase compared to the pilot study. **I shared knowledge and experience**: Only 38.9% agreed that they were able to share knowledge and experience, also a decrease compared to the pilot study.

As shown in Table 4, there were no statistically significant differences between barcamps.

### Qualitative results

Qualitative data are summarized in Table 5. 50% of the participants in both studies appreciate the opportunity for intensive discussions and exchanges. The variety of new topics and ideas introduced due to the event's openness was reported as beneficial by 80% of respondents in the pilot study and 50% in the feasibility study. Similar values were attributed to the extraordinary informal atmosphere conducive for networking.

The areas flagged for improvement presented a more varied picture. The allocation of time for instructions or input was contested; it was considered excessive by some and not sufficient by others. The need for more structure in the topic plan was emphasized more in the feasibility study, cited by 44% of the participants compared to only 10% in the pilot study. Concerns related to the sufficiency of the working environment for the sessions were echoed at both barcamps. Almost a third of the participants in the pilot study criticized the too small number of participants. Further suggestions included a more structured discussion during sessions, improving the documentation of results at the end of a session, and the wish for a whole-day barcamp at the upcoming GMA meeting in 2024.

Regarding suggestions for future barcamp topics, there was a shift in focus between the two studies. In the pilot study, networking among participants and digital teaching were highlighted by 20% of respondents. In the feasibility study, artificial intelligence and future trends in education attracted the interest of the participants.

## Discussion

### General overview

Our mixed-methods study aims to contribute to the efficacy and feasibility of barcamps as part of conferences in the field of medical education [4–6]. The pedagogical model of barcamps,

**Table 4. Mann-Whitney U Test results comparing pilot and feasibility studies.**

| Metric | Pilot 2022 | Feasibility 2023 | P-Value |
| --- | --- | --- | --- |
| **I learned from others** | 90% | 94.4% | 0.707 |
| **I networked with others** | 70% | 33.3% | 0.071 |
| **It was entertaining and fun** | 80% | 66.7% | 0.481 |
| **I found a way into the topic** | 30% | 50% | 0.328 |
| **I shared knowledge and experience** | 70% | 38.9% | 0.128 |

**Table 5. Qualitative data.**

| Question | Pilot study (2022) n = 10 | Feasibility study (2023) n = 18 |
|---|---|---|
| **What were the positive aspects of the barcamp?** | 5 x opportunities for intensive discussions/exchange | 9 x opportunity for intensive discussions/exchange |
| | 8 x variety of new topics/ideas due to openness | 8 x variety of new topics/ideas due to openness |
| | 8 x extraordinary informal atmosphere for networking | 8 x extraordinary informal atmosphere for networking |
| | | 1 x everything |
| **What was missing and what could be improved at the barcamp?** | 1 x too much time for instructions/input | 2 x too much time for instructions/input |
| | | 4 x not enough instructions/input |
| | 1 x more structure in establishing the topics plan | 8 x more structure in establishing the topics plan |
| | 1 x not sufficient working environment for the sessions | 2 x not sufficient working environment for the sessions |
| | 3 x too few participants/topics | 6 x more time for sessions |
| | 1 x improving the documentation of results at the end of the session | 2 x more structured discussions |
| | 1 x I have not yet understood the barcamp principle | 1 x plan a whole day for GMA2024 meeting |
| **For the next barcamp, I would appreciate the following topics** | 2 x networking among participants | 2 x artificial intelligence |
| | 2 x digital teaching | 2 x future trends in education |
| | 1 x open educational resources | 1 x future trends in examinations |
| | 1 x content or learning management systems | 1 x clinical rotations |
| | 1 x ethics of digital medicine | 1 x create seminar concepts together |
| | | 1 x topics should always be chosen short term u current |
| | | 1 x no matter |

which eschews traditional top-down information dissemination in favor of a participant-driven approach, has been previously discussed in the literature as a promising avenue for fostering engagement and active learning [2, 3]. With two separate evaluations, a pilot study in 2022 and a feasibility study in 2023, the study sheds light on participant evaluations, topic selections, and the likelihood of attending future events.

## Participants

Due to the general setting of the GMA conferences, we were unable to pursue any recruitment strategies. Unfortunately, we also did not evaluate the reasons or willingness of participants to participate. Therefore, we can't rule out a self-selection bias, in which individuals disproportionately select themselves into the group of participants [14]. Therefore, generalizations of our results should be made with caution. A general positive attitude or curiosity towards the learning format of a barcamp must be assumed. On the other hand, we cover a wide range of ages and genders, so there is no obvious bias from GMA conference attendees in general. In terms of personal characteristics, participants are most likely to be open to various new topics or ideas, to new methods and techniques, to support peer-to-peer methods and self-education, and to prefer low hierarchies with an extraordinarily informal atmosphere for networking [3].

## Participant evaluations and future attendance

Both studies showed high rates of participant evaluations, although the pilot study had a notably higher rate (91%) compared to the feasibility study (53%). The novelty of the concept in 2022 could have been a motivating factor for the evaluations.

The general favorability for the barcamp model, indicated by mean scores greater than 4 on a scale of 5, is consistent with the literature suggesting that barcamps offer a positive learning environment [6].

One of the most consistent findings across both evaluations was the participants' appreciation for the opportunities for intensive discussions and exchanges. This is consistent with the constructivist theories of learning (e.g. the interactionist constructivism of Reich [15]), which emphasize the importance of social interaction and dialogue in the learning process.

In the smaller pilot group, active participation during the barcamp appeared to be higher, as indicated by 70% being able to network and share knowledge and experience. In the larger group of the feasibility study, only one third were able to do so. In this group, open feedback indicated the desire for more time, showing that the number of participants is likely for be relevant to the success of this educational method. To deal with larger learning groups impeding the contribution of all participants, more time for the barcamp could be planned or groups could be divided into two or three, all discussing the same topic. Furthermore, in the larger group only nine topics were pitched compared to 16 topics in the smaller pilot group. Strategies must be found to activate more participants to pitch topics, for example by using a think-pair-share method to find and collect topics from the beginning [16].

## Topic selection and collaboration

The variety of topics pitched and discussed during both studies underscores one of the key advantages of the barcamp model: its flexibility and adaptability. The range of topics, including those related to generative AI, also demonstrates the model's ability to respond to emerging trends and technologies in medical education. In other settings, barcamps have even been used to achieve organizational change [17]. The autonomy in choosing topics combined with social interaction and, in parts the perception of competence can have a positive influence on learning motivation [18].

## Interactive and collaborative tools

The participants engaged in discussions about the use of interactive and collaborative tools to document the results of a barcamp. Collaborative tools for barcamp documentation are used to document the sharing of resources, ideas, and challenges among attendees during session conversations. These tools enable participants to reflect on collective learning, evaluate the success of the discussion, and continue the conversation beyond the event itself [2]. Examples of such tools include shared online platforms such as, e.g. https://cryptpad.fr/ [13], Google documents or Wikis [3].

## Experience and environment

Participants had limited prior experience with barcamps. Interestingly, the experience was greater at the first barcamp. This was most probably due to very limited general knowledge about barcamps in the medical education community in the first place. Because of the low number of participants in 2022, the share of the organization committee was larger. Furthermore, in 2023 the participants of 2022 spread the word and potential participants of the GMA annual meeting were made aware of the barcamp through an editorial in the Journal of

Medical Education [10]. The lack of experience with barcamps in the 2023 group is consistent with the stated need of four participants for more instruction or input about the method. Even though two other stated there was too much time wasted for instructions, a proper introduction is needed in order to enable all participants can actively take part and benefit from the method. Providing information on barcamps in advance, with the instruction to ensure that the structure is understood, can support a better understanding of each participant. Still, there should be enough spare time to explain the method in order to increase active participation of the attendees.

For transfer to medical education the age and respective experience of the participants need to be considered. The practical experience might be less and the group of learners probably more homogenous than the participants in our study. It might be interesting to investigate how and what participants in earlier stages of education discuss compared to those in higher semesters or clinical and / or educational staff.

### Areas for improvement and future research

The study revealed areas for improvement, such as the allocation of time for instructions and the need for a more structured topic plan. The latter finding is particularly interesting, as it suggests a potential tension between the barcamp model's inherent flexibility and participants' desire for structure. This is consistent with previous research that has explored the balance between learner autonomy and guidance in educational settings [19].

Furthermore, concerns were raised about the sufficiency of the learning environment, echoing sentiments previously described for barcamps [6].

That only 30% of the pilot group and 50% in the feasibility group stated that they found a way into the topic could be a sign that more time is needed to effectively enable the participants to understand and engage with the special characteristics of barcamps. In terms of a constructivist approach the combination of cognitive and social construction [20] can be useful for competency development. Therefore, in terms of a flipped or inverted classroom model [21], researcher tasks could be given to the attendees in advance to construct cognitive knowledge construction. Within the consecutive barcamp session, this knowledge can be consolidated in social construction. This is also in line with the reported positive effects reported of combining constructive tasks and interactive learning opportunities within discussions [22]. Since the interactive aspect is crucial for barcamps, methods like this can also be useful for interprofessional education to allow learners with different professional backgrounds to learn about, from and with each other [23]. As one of the benefits of a barcamp is the equal status of all participants and low hierarchies, it can also serve as a strategy enabling interprofessional socialization in terms of reducing barriers and working towards common goals [24].

### The use of barcamp principals in other medical education formats

The principals of barcamps might not only be applied to unconferences concerning medical education. We propose that the versatile, participant-driven approaches might also be applied to other formats like lectures, seminars, and courses, leading to un-lectures, un-seminars or un-courses. Joe Ruhl's six Cs: choice, collaboration, communication, critical thinking, creativity, and caring [8] can all be implemented particularly well.

The opportunity for intensive discussions/exchange, the openness for diversity of new topics/ideas, and an extraordinary informal atmosphere for networking might have positive impacts on learning processes.

Specific ideas for the use of barcamp principles in medical curricula might include but are not limited to the following.

**Un-problem based learning.**    During problem-based learning (PBL), the interactive steps might be replaced by pitches.

**Un-inverted classroom model.**    For the inverted classroom model, the barcamp approach could be used to allow students to define their own learning objectives or learning methods, e.g. with a pitch at the beginning of the presence phase.

**Un-lecture.**    After a short keynote impulse by lecturers, students could pitch specific sub-topics and work in small subgroups of up to 8 students in buzz-groups.

**Un-seminar and un-course.**    After lecturers propose some input and delineate learning objectives, students pitch different learning methods, e.g. whether they do a web-search, produce a podcast or shoot a short instructional video, to name only a few. This would enable students to activate previous knowledge and take into account personal interests.

**To be considered for un-learning environments.**    We propose that including the students in the decision of choosing their own learning method and setting priorities concerning sub-topics will lead to an extraordinary informal, motivating atmosphere for interactive networking and learning.

Interactive and collaborative tools for documenting lessons learned might improve the learning success as such and support consolidation of learning content.

This new approach poses particular challenges that need to be taken into account. One would have to consider that pitching and documentation of sessions should not be too time-consuming. For lecturers, it will be a challenge to provide structure and guidance. For the documentation of the sessions, a suitable learning management system might be needed. An e-Portfolio might prove useful in this regard.

Despite the complexity and revolutionary new approach, the results of the study show that the implementation is possible even with little experience of barcamps.

## Limitations

Several limitations in this study should be acknowledged to better interpret the findings:

1. Sample size: The most significant limitation is the small sample size in both the pilot and feasibility studies. Small samples can introduce bias and limit the generalizability of the results.

2. Statistical power: The limited sample size also affects statistical power, potentially fail to detect real differences or relationships. Thus, the lack of statistically significant results should be interpreted with caution.

3. Self-selection bias: The participants in these studies were attendees of the GMA conferences, which may introduce a self-selection bias, as these individuals may already have a favorable view of such educational formats. The selection-bias might reduce the reliability and generalizability of our findings.

4. Measurement tools: Although Likert scales are commonly used for educational evaluations, they have their own set of limitations, including the potential for central tendency bias and social desirability bias.

By acknowledging these limitations, we hope to provide a balanced view of the study's contributions and set the stage for future research that can address these gaps. However, to actually prove the efficacy or superiority of barcamps in learning, one would have to compare the learning outcome by comparing other collaborative methods, such as PBL or seminars, to a barcamp regarding the same topic and a subsequent exam to determine what has actually been learned to what extent.

## Conclusions

The article provides a contribution to the limited but growing body of literature on the use of barcamps in medical education [4–6]. Its findings suggest that this model holds promise as a versatile, participant-driven approach to learning. However, like any educational intervention, its efficacy is contingent on a variety of factors, including participant engagement, logistical support, and alignment with broader curricular goals.

The study's results demonstrate high levels of participant satisfaction and engagement, reinforcing the potential of barcamps as a viable educational tool to revitalize conferences. The most beneficial aspect of the barcamps perceived was that participants learned from others. The diversity in topics pitched and discussed further emphasizes the model's adaptability to various educational needs and emerging trends in medical education. However, the study also identified areas requiring improvement, particularly in relation to the structure of topic plans and the overall learning environment.

The opportunity to pitch learning methods and set their own priorities about subtopics might be especially rewarding for students. Barcamps could be superior in fostering engagement, practical skills, and collaborative learning, especially for subjects that benefit from discussion and peer interaction.

Extending the benefits of barcamps to other formats such as lectures, seminars, PBL and even inverted classroom might improve motivation and the learning process with more interaction, choice, collaboration, communication, critical thinking, creativity and last but not least caring about each other.

## Author Contributions

**Conceptualization:** Bernd F. M. Romeike, Daniel Tolks.

**Data curation:** Bernd F. M. Romeike, Christoph Stosch, Sören Moritz, Martin Lemos, Johanna Mink, Daniel Tolks.

**Formal analysis:** Bernd F. M. Romeike, Christoph Stosch, Sören Moritz, Daniel Tolks.

**Investigation:** Bernd F. M. Romeike, Christoph Stosch, Sören Moritz, Marianne Behrends, Daniel Tolks.

**Methodology:** Bernd F. M. Romeike, Marianne Behrends, Daniel Tolks.

**Project administration:** Bernd F. M. Romeike, Johannes Lang, Daniel Tolks.

**Resources:** Bernd F. M. Romeike.

**Supervision:** Johannes Lang, Sören Moritz, Marianne Behrends.

**Validation:** Bernd F. M. Romeike, Christoph Stosch, Sören Moritz, Daniel Tolks.

**Visualization:** Bernd F. M. Romeike.

**Writing – original draft:** Bernd F. M. Romeike, Johannes Lang, Christoph Stosch, Sören Moritz, Marianne Behrends, Martin Lemos, Johanna Mink, Daniel Tolks.

**Writing – review & editing:** Bernd F. M. Romeike, Johannes Lang, Christoph Stosch, Sören Moritz, Marianne Behrends, Martin Lemos, Johanna Mink, Daniel Tolks.

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
