## [Decision Letter · Decision Letter 0]

17 Apr 2024

PONE-D-24-01118Barcamps or Unconferences as an Emerging Paradigm in Medical Education: Insights from a Pilot and Feasibility Mixed Methods StudyPLOS ONE

Dear Dr. Romeike,

Thank you for submitting your manuscript to PLOS ONE. After careful consideration, we feel that it has merit but does not fully meet PLOS ONE’s publication criteria as it currently stands. Therefore, we invite you to submit a revised version of the manuscript that addresses the points raised during the review process.

We look forward to receiving your revised manuscript.

Kind regards,

Mukhtiar Baig, Ph.D.

Academic Editor

PLOS ONE

Journal Requirements:

Reviewers' comments:

Reviewer's Responses to Questions

**Comments to the Author**

1. Is the manuscript technically sound, and do the data support the conclusions?

Reviewer #1: Yes

Reviewer #2: Partly

2. Has the statistical analysis been performed appropriately and rigorously? 

Reviewer #1: Yes

Reviewer #2: Yes

3. Have the authors made all data underlying the findings in their manuscript fully available?

Reviewer #1: Yes

Reviewer #2: Yes

4. Is the manuscript presented in an intelligible fashion and written in standard English?

Reviewer #1: Yes

Reviewer #2: Yes

5. Review Comments to the Author

Reviewer #1: The submitted article is a mixed methods study investigating the introduction of the 'old and new' barcamp method into medical education. I believe that this is an important study that is highly original and could have significant implications for future medical education.

However, to aid the reader's understanding, I would like to ask for additional consideration of the following points.

1. Please describe in more detail how and why you and your colleagues decided to consider Barcamps.

2. The Barcampus was held as part of The annual conference of the DACH Association for Medical Education, please describe and tell us how you and your colleagues recruited participants. The results of the study may vary depending on the reasons and willingness of the participants to participate. This is an issue related to the self-selection bias described as a limitation.

3. If the study allows for discussion, please discuss what personal characteristics are suitable for Barcamps.

4. The word 'Unconference' is used along with 'Barcamps' in the title and conclusion of the paper, but the word 'Unconference' is rarely mentioned in the text. What is the difference between 'Barcamps' and 'Unconferences' and what is the significance of using the term 'Unconference' in the paper?

Reviewer #2: The manuscript “Barcamps or Unconferences as an Emerging Paradigm in Medical Education: Insights from a Pilot and a Feasibility Mixed Method Study” by Romeike et al. investigates the feasibility and efficacy of using barcamps as an educational tool in medical education. It uses a mixed-methods approach to offer a comprehensive view of the pedagogical potential and participant reception of barcamps in a medical educational setting. The study's relevance to the journal's focus on advancing medical education through innovative methods is clear and well-established.

Authors appropriately addressed ethical considerations by obtaining verbal consent from participants and justifying the lack of formal ethical review due to the low-risk nature of the research. Even though a formal ethical review or waiver would enhance credibility, authors’ choice aligns with norms in educational research.

The mixed-methods design is acceptable for the exploratory nature of the study and the study’s major limitations such as small sample size and self-selection bias are openly discussed. However, I believe that another important aspect should be discussed. Indeed, authors declare that the participants of the studies were attendees of the medical education barcamps held in 2022. This can introduce additional biases that may affect the reliability and generalizability of the study's findings, such as selection bias influencing the general positive disposition towards this learning format. I believe that clarifying participants’ recruitment criteria is therefore necessary, as it is including this point in the study’s limitations.

Finally, I believe that giving a wider introduction would help making the study more clear. Authors could perhaps move the paragraph “The BarCamps”, currently part of the materials and methods section, to the introduction and also discuss other forms of innovative teaching methods that are part of the paradigm shift that is happening in medical education. In this regard, I suggest reading two articles that could be useful: “Dissecting human anatomy learning process through anatomical education with augmented reality: AEducAR 2.0, an updated interdisciplinary study” (doi: 10.1002/ase.2389) and “Graphic medicine meets human anatomy: The potential role of comics in raising whole body donation awareness in Italy and beyond. A pilot study” (doi: 10.1002/ase.2232).

The manuscript is recommended for acceptance after minor revisions are addressed, ensuring that the presented research meets the highest standards of scientific and ethical integrity. Indeed, the presented study holds significant promise for informing and shaping future educational practices in the medical field.

6. PLOS authors have the option to publish the peer review history of their article (what does this mean?). If published, this will include your full peer review and any attached files.

Reviewer #1: No

Reviewer #2: No

---

## [Author Response · Author response to Decision Letter 0]

2 Jul 2024

Dear Mukhtiar Baig, dear reviewers.

Thank you, a lot, for your very valuable comments on our manuscript titled "Barcamps or Unconferences as an Emerging Paradigm in Medical Education: Insights from a Pilot and Feasibility Mixed Methods Study". We greatly appreciate the comments, that contribute to a significant improvement or our manuscript. 

Please find here our point-to-point reply to the reviewers’ comments.

Reviewer #1: 

The submitted article is a mixed methods study investigating the introduction of the 'old and new' barcamp method into medical education. I believe that this is an important study that is highly original and could have significant implications for future medical education.

However, to aid the reader's understanding, I would like to ask for additional consideration of the following points.

1. Please describe in more detail how and why you and your colleagues decided to consider Barcamps.

We now describe in greater detail how and why we decided to consider barcamps (lines 77-80 in Revised Manuscript with Track Changes). 

2. The Barcampus was held as part of The annual conference of the DACH Association for Medical Education, please describe and tell us how you and your colleagues recruited participants. The results of the study may vary depending on the reasons and willingness of the participants to participate. This is an issue related to the self-selection bias described as a limitation.

The recruitment of participants is now described in greater detail in the Materials and methods section (lines 127-129), in the Result section (lines 157-160), and it is discussed in the Discussion section (lines 264-274). Here, a new reference has been added ([14] lines 464-465). Finally, the limitations have been specified (lines 399-402). 

3. If the study allows for discussion, please discuss what personal characteristics are suitable for Barcamps.

This is a very interesting aspect, which we now address in the discussion section, including a citation that was already in the reference list (lines 271-274).

4. The word 'Unconference' is used along with 'Barcamps' in the title and conclusion of the paper, but the word 'Unconference' is rarely mentioned in the text. What is the difference between 'Barcamps' and 'Unconferences' and what is the significance of using the term 'Unconference' in the paper?

There is no difference between 'barcamps' and 'unconferences'. This is now stated in the Abstract (line 25) and in the Introduction (lines 57-59), with the addition of a new reverence ([1] lines 433-434). This precision is now followed by the transfer of the description of barcamps from the Materials section to the Introduction, as suggested by Reviewer #2 (lines 59-73). In other sections, the term "Unconferences" has been removed to avoid confusion (lines 413 and 428). It has been retained in the section "The use of barcamps in other medical education formats" (line 354) to clarify the wording of un-lectures, un-seminars or un-courses.

Reviewer #2: 

The manuscript “Barcamps or Unconferences as an Emerging Paradigm in Medical Education: Insights from a Pilot and a Feasibility Mixed Method Study” by Romeike et al. investigates the feasibility and efficacy of using barcamps as an educational tool in medical education. It uses a mixed-methods approach to offer a comprehensive view of the pedagogical potential and participant reception of barcamps in a medical educational setting. The study's relevance to the journal's focus on advancing medical education through innovative methods is clear and well-established.

Authors appropriately addressed ethical considerations by obtaining verbal consent from participants and justifying the lack of formal ethical review due to the low-risk nature of the research. Even though a formal ethical review or waiver would enhance credibility, authors’ choice aligns with norms in educational research.

The mixed-methods design is acceptable for the exploratory nature of the study and the study’s major limitations such as small sample size and self-selection bias are openly discussed. However, I believe that another important aspect should be discussed. Indeed, authors declare that the participants of the studies were attendees of the medical education barcamps held in 2022. This can introduce additional biases that may affect the reliability and generalizability of the study's findings, such as selection bias influencing the general positive disposition towards this learning format. I believe that clarifying participants’ recruitment criteria is therefore necessary, as it is including this point in the study’s limitations.

This concern was also raised by reviewer #1. As mentioned above, the recruitment of participants is now described in more detail in the Materials and methods section (lines 127-129 in the revised manuscript with track changes), in the Results section (lines 157-160), and is discussed in the Discussion section (lines 264-274). A new reference is added here ([14] lines 464-465). Finally, the limitations have been specified (lines 399-402). 

Finally, I believe that giving a wider introduction would help making the study more clear. Authors could perhaps move the paragraph “The BarCamps”, currently part of the materials and methods section, to the introduction… 

We have moved large parts of the "The barcamps" section from the Materials and methods section to the Introduction (lines 59-73).

… and also discuss other forms of innovative teaching methods that are part of the paradigm shift that is happening in medical education. In this regard, I suggest reading two articles that could be useful: “Dissecting human anatomy learning process through anatomical education with augmented reality: AEducAR 2.0, an updated interdisciplinary study” (doi: 10.1002/ase.2389 ) and “Graphic medicine meets human anatomy: The potential role of comics in raising whole body donation awareness in Italy and beyond. A pilot study” (doi: 10.1002/ase.2232) .

As for other forms of innovative teaching methods that are part of the paradigm shift in medical education, we had a tough discussion among the authors of this manuscript. 

The two articles mentioned are very well written and describe very interesting future innovative educational practices. However, the focus of our own manuscript is more on medical education meetings and not so much on the whole spectrum of teaching methods, which is clearly beyond the scope of our article. It is only at the end of our article that we consider changing the format of medical education. In the end, we decided that the two articles described were not very appropriate for our article. 

The manuscript is recommended for acceptance after minor revisions are addressed, ensuring that the presented research meets the highest standards of scientific and ethical integrity. Indeed, the presented study holds significant promise for informing and shaping future educational practices in the medical field.

Thank you very much for this estimation.

In addition, we have changed some words for better readability or understanding:

- Line 26: using instead of utilizing

- New heading line 352-3: The use of barcamps in other medical education formats

- Line 354 unconferences instead of un-conferences

- Addition in line 356: other formats like

In accordance with the instructions for authors 

- Capital letters are used in the address of all authors (line 19)

- all headings have been changed to lower case

- all references were updated.

@ Paula Katrina A. Maderazo:

Dear Dr. Romeike,

We've checked your submission and before we can proceed, we need you to address the following issues:

1. "We note that your Data Availability Statement is currently as follows: [All relevant data are within the manuscript and its Supporting Information files]

Dear Paula Katrina A. Maderazo,

Thank you for your message and for providing detailed guidelines on the data availability requirements for our submission to PLOS ONE. We understand the importance of sharing the "minimal data set" as defined by the journal, which includes the values behind the reported statistical measures.

However, after careful consideration and discussion among all co-authors, we have decided not to provide additional data beyond what is presented in the manuscript. We believe that the data included in our submission is sufficient to understand and replicate the study findings, and that providing further data would not significantly contribute to the paper.

More importantly, we have serious concerns regarding the privacy and confidentiality of our study participants.Due to the small sample size of our pilot and feasibility study, there is a risk that individuals could be accidentally identified, despite our best efforts to de-identify the data. This could endanger the personal rights of our subjects, which is something we must avoid at all costs.

As a compromise, we propose the following addition to the data availability section:

The complete datasets cannot be shared publicly because we have taken great care to maintain the confidentiality of our participants' data. Sharing the data in a public repository, even in a de-identified 

format, could still pose a risk to participant privacy, especially since our dataset is small and the possibility of re-identification is not negligible. However, The datasets of the current study are available from the corresponding author for researchers who meet the criteria for access to confidential data.

We hope you understand our position and the ethical considerations that have led us to this decision. We are willing to discuss alternative solutions that would satisfy the journal's requirements without compromising the privacy and confidentiality of our study participants.

2. We note that several of your files are duplicated on your submission. Please remove any unnecessary or old files from your revision, and make sure that only those relevant to the current version of the manuscript are included.

The duplicated files are removed.

We believe that our revised manuscript will be of great interest to the readers of PLOS ONE, given its relevance and contribution to the field. We look forward to the possibility of sharing our work with your audience and eagerly await your response.

Best regards,

Bernd Romeike for all authors

---

## [Editor Report · Decision Letter 1]

6 Aug 2024

Barcamps or Unconferences as an Emerging Paradigm in Medical Education: Insights from a Pilot and Feasibility Mixed Methods Study

PONE-D-24-01118R1

Dear Dr. Romeike,

We’re pleased to inform you that your manuscript has been judged scientifically suitable for publication and will be formally accepted for publication once it meets all outstanding technical requirements.

Kind regards,

Mukhtiar Baig, Ph.D.

Academic Editor

PLOS ONE

---

## [Editor Report · Acceptance letter]

8 Aug 2024

PONE-D-24-01118R1 

PLOS ONE

Dear Dr. Romeike, 

I'm pleased to inform you that your manuscript has been deemed suitable for publication in PLOS ONE. Congratulations! Your manuscript is now being handed over to our production team.

Kind regards, 

on behalf of

Professor Mukhtiar Baig 

Academic Editor

PLOS ONE